# Changes in Memory, Sedation, and Receptor Kinetics Imparted by the β2-N265M and β3-N265M GABA_A_ Receptor Point Mutations

**DOI:** 10.3390/ijms24065637

**Published:** 2023-03-15

**Authors:** Alifayaz Abdulzahir, Steven Klein, Chong Lor, Mark G. Perkins, Alyssa Frelka, Robert A. Pearce

**Affiliations:** Department of Anesthesiology, University Wisconsin, Madison, WI 53705, USA; abdulzahir@wisc.edu (A.A.);

**Keywords:** anesthetic action, general anesthesia, amnesia, GABA, intravenous anesthetics, contextual memory, hippocampus

## Abstract

Point mutations in the β2 (N265S) and β3 (N265M) subunits of γ-amino butyric acid type A receptors (GABA_A_Rs) that render them insensitive to the general anesthetics etomidate and propofol have been used to link modulation of β2-GABA_A_Rs to sedation and β3-GABA_A_Rs to surgical immobility. These mutations also alter GABA sensitivity, and mice carrying the β3-N265M mutation have been reported to have impaired baseline memory. Here, we tested the effects of the β2-N265M and β3-N265M mutations on memory, movement, hotplate sensitivity, anxiety, etomidate-induced sedation, and intrinsic kinetics. We found that both β2-N265M and β3-N265M mice exhibited baseline deficits in the Context Preexposure Facilitation Effect learning paradigm. Exploratory activity was slightly greater in β2-N265M mice, but there were no changes in either genotype in anxiety or hotplate sensitivity. β2-N265M mice were highly resistant to etomidate-induced sedation, and heterozygous mice were partially resistant. In rapid solution exchange experiments, both mutations accelerated deactivation two- to three-fold compared to wild type receptors and prevented modulation by etomidate. This degree of change in the receptor deactivation rate is comparable to that produced by an amnestic dose of etomidate but in the opposite direction, indicating that intrinsic characteristics of GABA_A_Rs are optimally tuned under baseline conditions to support mnemonic function.

## 1. Introduction

Many drugs, including benzodiazepines, barbiturates, neurosteroids, and the intravenous general anesthetics propofol and etomidate, act as positive allosteric modulators of γ-aminobutyric acid type A receptors (GABA_A_Rs) [1,2]. They produce a wide variety of effects, ranging from anxiolysis, sedation, and memory impairment at low doses to hypnosis, respiratory depression, and surgical immobility at higher doses. The spectrum of effects produced by a given drug is determined by both the dose that is administered and the subtype of GABA_A_R that is targeted [3,4,5]. Elucidating the mechanisms by which anesthetics produce their desired effects, and undesired side-effects, remains an important research goal.

GABA_A_Rs are heteropentameric ligand-gated ion channels that collectively comprise the major class of inhibitory receptors in the mammalian brain [6]. Each receptor is formed by five structurally similar transmembrane subunits that surround a central chloride-permeable ion pore. Each subunit exists in multiple isoforms (α1-6, β1-3, γ1-3, δ, ε, θ, and ρ1-3), with the majority of GABA_A_Rs composed of two α, two β, and one γ subunit [7]. Although millions of possible subunit combinations exist, it has been estimated that only about 25 different subunit combinations are present in the mammalian brain [8]. The different receptor subtypes display distinct physiological properties and pharmacological sensitivities, and their expression levels depend on brain region, cell type, and even subcellular location [9].

To link drug effects at the behavioral level to modulation of specific GABA_A_R subtypes, a particularly fruitful approach has utilized mice carrying mutations that make them insensitive to specific drugs. GABA_A_R α-subunits have been studied most extensively. Taking advantage of single point mutations in the extracellular domain that make receptors insensitive to benzodiazepines [10], individual receptor subtypes have been linked to specific behavioral characteristics, e.g., α1 to sedation and amnesia [10], α2 and α3 to anxiety [11,12,13], and α5 to memory and cognition [14,15]. Similarly, for β-subunits, single point mutations in the transmembrane domain that make receptors insensitive to propofol and etomidate [16,17] have been used to link the modulation of β2-GABA_A_Rs to sedation and ataxia [18], and β3-GABA_A_Rs to the loss of righting reflex, respiratory depression, and loss of the hindlimb-withdrawal reflex [19,20]. With regard to memory, since etomidate modulates only GABA_A_Rs that incorporate β2 or β3 subunits [16], its ability to impair contextual conditioning in mice carrying the anesthetic-insensitive β3-N265M mutation indirectly implicates β2-GABA_A_Rs in memory suppression by etomidate [21]. Experiments *in vitro* in which etomidate blocked long-term potentiation in hippocampal brain slices from β3-N265M [21] but not β2-N265M mice [22] further support a role for β2-GABA_A_Rs, but not β3-GABA_A_Rs, in etomidate-induced memory suppression.

Curiously, despite evidence against the modulation of β3-GABA_A_Rs as a mechanism underlying anesthetic suppression of memory by etomidate, propofol, or isoflurane [21,23,24], other findings indicate that β3-GABA_A_Rs do play a role in memory formation and hippocampal network activity under control conditions. Specifically, β3-N265M mice were found to be deficient in contextual fear conditioning under drug-free conditions [23], but unchanged in several other behavioral characteristics, including open field activity, hotplate sensitivity, or passive avoidance learning or retention [20,24]. Moreover, in experiments *in vitro*, the strength of feedforward inhibition in the CA1 region of the hippocampus in brain slices from β3-N265M mice was reduced under control conditions [22], but feedback inhibition in the hippocampus [22] and spontaneous activity patterns in the somatosensory cortex [25] were unchanged. These relatively discrete changes in learning and in circuit properties in β3-N265M mice presumably reflect alterations in intrinsic receptor characteristics imparted by the point mutation, which have previously been shown to produce a slight rightward shift (2–3 fold) in the GABA dose-response relationship [26,27]. A rightward shift in GABA sensitivity of recombinant α1β2(N265M)γ2 receptors has also been reported [28], raising the possibility that β2-N265M mice might also exhibit changes in baseline behavioral characteristics.

The goals of the present study were (i) to assess the effects of the β2-N265M and β3-N265M mutations on several ‘baseline’ (drug-free) behavioral characteristics, including hotplate sensitivity, elevated plus maze exploration, and contextual fear conditioning; (ii) to compare the ability of β2-N265M versus β3-N265M mice to resist etomidate-induced sedation; (iii) to evaluate the effects of the β2-N265M mutation on spontaneous exploration under drug-free conditions and on recovery following dose-dependent suppression by etomidate; and (iv) to test the effects of the β2-N265M and β3-N265M mutations on intrinsic receptor kinetics and their modulation by etomidate. We found that (i) mice carrying the N265M mutation in either the β2 or β3 subunit exhibited impaired contextual conditioning, but they were unchanged in the other behavioral measures; (ii) β2-N265M mice, but not β3-N265M mice, resisted etomidate-induced sedation; (iii) the β2-N265M mutation produced a slight increase in exploratory activity under baseline conditions, but it strongly accelerated recovery from etomidate administered at doses up to 20 mg/kg IP (the highest we tested), in a gene dose-dependent manner, i.e., heterozygous mice showed a partial resistance; and (iv) the β2-N265M and β3-N265M mutations both accelerated receptor deactivation by approximately two- to three-fold following a brief pulse of GABA. Interestingly, this degree of change in receptor deactivation rate is comparable to that produced by an amnestic dose of etomidate but in the opposite direction, indicating that intrinsic characteristics of GABA_A_Rs are optimally tuned under baseline conditions to support mnemonic function.

## 2. Results

### 2.1. Behavioral Characteristics of β2-N265M and β3-N265M Mice 

We carried out behavioral studies in mixed-sex groups of 15–23 mice for Context Preexposure Facilitation Effect (CPFE) studies, or 6–15 mice for Open Field Test (OFT), hotplate test, and Elevated Plus Maze (EPM) studies. We compared mice carrying either the β2-N265M or the β3-N265M mutation on a mixed C57 × 129 background versus their wild type (WT) littermates. 

#### 2.1.1. CPFE Learning Test

We first measured the effect of the β2-N265M and β3-N265M mutations on contextual memory using the CPFE paradigm. We found that WT mice from both lines exhibited substantial freezing behavior on day 3 (Figure 1A left, β2-WT 33.9 ± 7%; Figure 1A right, β3-WT 42.8 ± 7%), demonstrating that they recalled the test chamber in which they had received a shock on day 2. Surprisingly, mutant mice from both lines spent significantly less time freezing on day 3 compared to their WT counterparts (Figure 1A left, β2-N265M 11.9 ± 3.4%, *p* = 0.04 vs. β2-WT; Figure 1A right, β3-N265M 11.6 ± 3.8%, *p* = 0.002 vs. β3-WT). This finding indicates that both mutations caused an impairment of baseline memory. 

#### 2.1.2. Hotplate Test

The hotplate test provides information about pain sensitivity. This characteristic is particularly important in the present setting because any differences in nociception imparted by the mutation could alter the effectiveness of the shock administered during the conditioning phase of the CPFE test. We found no significant differences between the WT, β2-N265M, and β3-N265M mice (Figure 1B; β2-WT 6 ± 1 s, β2-N265M 7 ± 1 s, *p* = 0.57; β3-WT 7 ± 0.5 s, β3-N265M 6 ± 1 s, *p* = 0.25).

#### 2.1.3. Elevated Plus Maze Test

The EPM test provides information about anxiety. We found no significant differences between the WT, β2-N265M, and β3-N265M mice in number of open arm entries (Figure 1(Ci); β2-WT 10 ± 4%, β2-N265M 7 ± 2%, *p* = 1; β3-WT 18 ± 6 s, β3-N265M 16 ± 6%, *p* = 0.79) or in time spent on open arms (Figure 1(Cii); β2-WT 25 ± 6%, β3-N265M 31 ± 4%, *p* = 0.43; β3-WT 6 ± 2%, β3-N265M 4 ± 1%, *p* = 0.66).

#### 2.1.4. Sedation

We compared the effects of the β2-N265M and β3-N265M mutations on etomidate-induced sedation by injecting either saline (control) or etomidate (9 mg/kg *i.p.*) 30 min before placing in the same chamber used for CPFE studies. We found that in the saline treatment group, WT mice from both lines spent only ~5% of the time immobile, i.e., 95% of the time exploring the chamber (Figure 2A, sal-β2-WT; Figure 2B, sal-β3-WT). WT mice administered etomidate spent approximately 80% of the time immobile (Figure 2A, sal-β2-WT; Figure 2B, sal-β3-WT), showing that this dose of etomidate was strongly sedative. Mutant mice from both lines in the saline treatment group also spent more than 95% of the time exploring the chamber (Figure 2A, sal-β2-N265M; Figure 2B, sal-β3-N265M), similar to their WT counterparts. As previously reported [18], β2-N265M in the etomidate treatment group were highly resistant to the sedative effects of etomidate, and they continued to explore the chamber at a level that was not significantly different than the saline treatment group (Figure 2A, sal-β2-N265M 0.9 ± 0.2%, etom-β2-N265M 3.5 ± 1.1%, *p* = 0.2). By contrast, β3-N265M mice were not resistant to the sedative effect of etomidate, and they spent much of the time immobile (etom-β3-N265M 63.5 ± 5.5%), similar to the etomidate treatment group in β3-WT mice.

We further investigated the degree of resistance that the β2-N265M mutation provides to the sedative effect of etomidate by measuring the recovery of locomotor activity in an open field following various doses of etomidate (10 mg/kg, 15 mg/kg, and 20 mg/kg). Under drug-free conditions, the β2-N265M mutant mice had slightly but significantly higher baseline locomotor activity than the β2-WT or heterozygous β2-N265M mice (β2-Het) (Figure 3A, β2-WT 1122 ± 112 cm, β2-Het 1007 ± 173 cm, β2-N265M 1601 ± 136.5 cm, β2-WT vs. β2-N265M, *p* = 0.0017; β2-het vs. β2-WT, *p* < 0.001, one-way ANOVA). Following etomidate 10 mg/kg *i.p.*, β2-N265M mice regained exploratory activity within 10 min of the injection, whereas the β2-WT mice remained sedated for 30 min (Figure 3B). β2-Het mice recovered slightly faster than the β2-WT mice (Figure 3B, *p* = 0.029), demonstrating that resistance to sedation exhibits gene dose-dependence. Following etomidate 15 mg/kg and 20 mg/kg *i.p.*, only β2-N265M mice recovered mobility within the 50 min duration of the test, and their recovery required progressively more time as the dose increased (Figure 3C,D). These results indicate that the β2-N265M mutation provides a strong resistance, but not complete insensitivity, to the sedative effect of etomidate.

### 2.2. Expressed Receptor Characteristics

The memory impairment and baseline change in mobility of the mutant mice prompted us to examine the impact of the mutation on intrinsic receptor characteristics. Prior studies of the N265M mutation focused on receptors in which mutant subunits were partnered with α1/2/3-GABA_A_Rs [26,28,29]. However, in the hippocampus, a high proportion of receptors incorporate α5 subunits, and a strong body of evidence ties α5-GABA_A_Rs to hippocampus-dependent memory and its disruption by etomidate [30]. Therefore, we examined the impact of the β-N265M mutation in combination with α5 and γ2L subunits on properties of recombinant receptors. 

We first measured the effect of the β2-N265M mutation on the GABA concentration-response relationship using a multi-barrel solution application system to apply pulses of GABA ranging from 1 μM to 3 mM, for 500 ms in duration, to whole-cell recordings of recombinant α5β2γ2L and α5β2(N265M)γ2L receptors expressed in HEK293 cells (Figure 4A). Similar to the effect of the mutation in α5β3γ2 receptors [27], we found that the mutation caused a rightward shift, slightly less than two-fold (Figure 4A, *p* < 0.001), with EC50 = 18.6 µM (WT) and 30.5 μM (β2-N265M).

To simulate the responses of receptors at synaptic or perisynaptic locations to brief transients of GABA, we used rapid solution exchange methods applied to outside-out patches from HEK293 cells expressing recombinant α5β2γ2L, α5β2(N265M)γ2L, α5β3γ2L, and α5β3(N265M)γ2L receptors. The effect of the mutation was similar for both β2- and β3-containing receptors: in the absence of etomidate, mutant receptors deactivated more rapidly than β2-WT and β3-WT receptors (Figure 4B; Table 1). The β2-N265M mutation also slowed activation compared to β2-WT receptors, and for β3-N265M receptors, there was a trend toward slower activation (Table 1). Biexponential curve fits of the deactivation phase revealed that both the fast and slow time constants were smaller for mutant receptors, and there was a shift of power toward the fast component. The net effect was that the weighted time constant of decay (τ_weighted_) of β2-N265M receptors was approximately one-half that of β2-WT (β2-WT 107 ± 10 ms, β2-N265M 60 ± 7 ms), and for β3-N265M receptors, it was one-third of β3-WT (β3-WT 114 ± 27 ms, β3-N265M 37 ± 10 ms).

Finally, we examined the effects of the N265M mutation on pharmacological modulation by etomidate (Figure 4C–F; Table 1). Etomidate (1 μM) slowed the deactivation of β2- and β3-WT receptors by approximately 2–3-fold (β2: τ_weighted_ etom/ctrl = 268 ± 14%, *p* < 0.001; β3: τ_weighted_ etom/ctrl = 214 ± 17%, *p* < 0.001), but it had no effect on mutant receptors (β2-N265M: etom/ctrl = 112 ± 13%, *p* = 0.47; β3-N265M: etom/ctrl = 102 ± 3%, *p* = 0.93).

## 3. Discussion

The major findings from this study are: (i) mice carrying either the β2-N265M or the β3-N265M mutation exhibit a deficiency in baseline memory performance in a hippocampus-dependent task; (ii) β2-N265M mice display increased exploratory activity under control conditions; (iii) β2-N265M mice resist the sedative effects of etomidate, but β3-N265M mice do not; (iv) the resistance of β2-N265M mice to sedation is gene dose-dependent, i.e., heterozygous mice are partially resistant; and (v) the N265M mutation in either the β2 or β3 subunit slows receptor activation and speeds deactivation following a brief pulse of GABA.

The changes in baseline behavioral characteristics imparted by the N265M mutation are unusual, but not without precedent. A multitude of behavioral and electrophysiological studies have previously found the β2-N265S and β3-N265M mutations to be ‘silent’, i.e., behavioral or physiological characteristics were not different in WT vs. mutant mice under drug-free conditions. These studies included motor activity [18,19], respiration [19], contextual fear conditioning [21], and whole-cell recordings of inhibitory synapses [18,31,32]. Similarly, in the present study, we also found no differences between WT and mutant animals in thermal sensitivity or anxiety (Figure 2). However, the memory deficit we observed does match the reduced baseline freezing scores for β3-N265M mice that were previously reported [23]. Since that study utilized a slightly different learning paradigm (fear conditioning to context and tone) and a different background strain (129/SvJ × 129/Sv), the conclusion that the β3-N265M mutation adversely influences baseline memory appears to be robust. The findings that β2-N265M mice also exhibit a memory deficit (Figure 1A) and display increased exploratory activity under drug-free conditions (Figure 3A) are novel observations, as is the finding that there is a gene dose-dependence of resistance to sedation (Figure 3B).

Since the β2-N265 mouse model studied previously utilized the β2-N265S mutation [18], as opposed to the β2-N265M mutation that we used here, it is worth considering whether any differences might be expected, or whether the two mutations are essentially equivalent. One major difference at the molecular level is that the β2-N265M mutation completely removes sensitivity to etomidate, whereas the β2-N265S mutation makes the receptor only partially insensitive [29]. Another difference is that the β2-N265M mutation caused a significant shift in GABA sensitivity manifested as a shift in the dose-response curve (Figure 4), whereas β2-N265S imparted little change [29]. Either of these differences could quite possibly influence the cellular, network, and behavioral properties of mice that carry the mutation and their response to etomidate. However, the resistance to sedation that we observed in β2-N265M mice (Figure 3) replicates the same finding in β2-N265S mice [18], demonstrating reproducibility and adding confidence to the conclusion that etomidate produces sedation through β2-GABA_A_Rs. 

These findings fit within an emerging pattern of site-specific and agent-specific anesthetic mechanisms. Shortly after the first public demonstrations of general anesthesia, investigators have sought to understand its scientific basis [33]. The long-standing quest for a unitary mechanism that would apply to all drugs and all manifestations has now turned to systematic investigations linking specific targets to specific ‘endpoints’ that are considered essential features of general anesthesia [3,34]. These include sedation and hypnosis, immobility in the face of a noxious stimulus, and amnesia for the event [1]. Among the many findings that have supported the so-called ‘Multisite Agent Specific’ model [35], perhaps the strongest support has come from experiments utilizing mice carrying the GABA_A_R β2/3-N265 point mutations. A mutation in the β2 subunit was found to impart resistance to sedation [18]—a key result that we confirm here (Figure 3)—whereas the mutation in the β3 subunit prevented surgical immobility [20]. Both β2- and β3- receptors were shown to contribute to ‘hypnosis’ (loss of righting reflex) [18,20,36]. These studies provided strong evidence that specific and distinct but overlapping molecular targets underlie these endpoints of anesthesia. By contrast, the anesthetic targets for impairing memory have remained largely undefined. Here, using a hippocampus-dependent contextual memory task, we showed that changes in either β2- or β3-GABA_A_R function can alter mnemonic function. However, since we did not measure resistance to etomidate suppression of memory, as both lines of mice exhibited baseline memory deficits, future studies utilizing a modified learning paradigm that does permit robust learning in these mutant strains will be required to directly assess the contributions of β2- and β3-GABA_A_Rs to memory suppression by etomidate and other GABAergic agents.

Both the β2-N265M mutation (Figure 4) and the β3-N265M mutation [27] imparted rightward shifts in the steady-state GABA concentration-response curves, as other investigators have similarly reported for other subunit combinations [26,28,29,37]. However, synaptic receptors are activated by transient pulses of GABA. Therefore, we went on to examine the effects of the mutation on receptor kinetics, using rapid solution exchange methods to mimic their activation *in situ*. Our finding that the mutation speeds deactivation (Figure 4) differs from a previous investigation of α1β2(N265S)γ2L and α1β2(N265M)γ2L receptors using rapid solution exchange methods [29], perhaps due to receptor subunit composition. However, the accelerated deactivation is generally consistent with the coupling that has been reported between agonist binding and channel gating [38,39]. 

Since both mutations lead to comparable changes in receptor kinetics, why do they not both confer resistance to sedation? The answer presumably relates to the differential expression of β2- versus β3-GABA_A_Rs in the cells and brain circuits that control the level of arousal. Although the precise targets remain undefined, several candidate regions in the rostral pons and preoptic area of the hypothalamus that control sleep and wakefulness have been implicated [1]. However, the complex circuitry in this region has made it difficult to establish mechanism of sedation and hypnosis at the molecular and cellular level. New methods that allow manipulation of specific sites and cells should provide additional insights to this important topic [40,41].

Interestingly, reduced agonist sensitivity and accelerated decay are opposite of the effects induced by a wide variety of anesthetics, including etomidate. Whereas the precise link between changes in GABA receptor function and memory remain undefined, it is noteworthy that both GABAergic hypofunction and hyperfunction lead to cognitive deficits that can be alleviated by negative and positive allosteric modulators of α5-GABA_A_Rs [42]. The U-shaped relationship between memory function and GABA_A_R properties fits well with the general concept that cellular properties are tuned to an optimum level to support circuit function [43]. 

Although the N265M mutation of either β2 or β3 subunit impairs memory, the underlying mechanisms might differ. β3 subunits are heavily concentrated in the dendritic regions where they partner with α5-GABA_A_Rs at GABA_A,slow_ synapses on pyramidal neurons to mediate long-lasting feedforward inhibition and a late component of feedback inhibition in the CA1 region of the hippocampus [22,44]. By contrast, β2 subunits are concentrated at the somatic layer [45], and they partner primarily with α1 subunits to mediate fast inhibitory postsynaptic currents in pyramidal neurons [46]. However, they also partner with α5 subunits to mediate an early component of long-lasting feedback inhibition [22]. In addition, β3 subunits are more heavily expressed by pyramidal neurons, whereas β2 subunits are concentrated in interneurons [45]. Interestingly, the elimination of α5-GABA_A_Rs from either pyramidal neurons or interneurons interferes with suppression of memory by low doses of etomidate [47], just as the N265M mutation of either β2- or β3-GABA_A_Rs interferes with memory formation under drug-free conditions. Thus, the intrinsic characteristics of both β2- and β3-GABA_A_Rs appear to be optimally tuned to support mnemonic function, and altering their activity by pharmacologic or genetic manipulations causes memory to fail. Identifying the specific circuit components that are most influential in these processes should illuminate mechanisms of anesthetic action and reveal crucial points of failure that might lead to memory dysfunction in neurodegenerative disorders associated with changes in GABAergic inhibition, such as Alzheimer’s Disease and other tauopathies [48,49].

## 4. Materials and Methods

### 4.1. Experimental Mice

The β2-N265M mice were generated on a C57BL/6J background utilizing CRISPR-Cas9 technology with procedures described previously [50]. Briefly, an *in vitro* transcribed gRNA with a target sequence (CCGGAGGTGGGTGTTGATTG) near the mutation site in Exon 9 of β2 was injected into C57BL/6J zygotes along with Cas9 mRNA and a 120-nucleotide single-stranded oligonucleotide repair template (IDT DNA, Coralville, IA, USA). A knock-in founder was screened with PCR and Sanger sequencing for mutations at the top 15 off-target sites predicted in silico, and identified mutations were eliminated from the pedigree following breeding with WT C57BL/6J mice.

Mice with the β2-N265M mutation in a mixed background were created by crossing homozygous β2-N265M (C57BL/6J) mice with WT 129X1/SvJ mice (Jackson Laboratories, Bar Harbor, Maine). This mating produced heterozygous mutant mice in a mixed background, which were bred together to create the experimental mice that we studied (i.e., β2-WT, and β2-N265M heterozygous and homozygous mutant littermates). Similarly, mice with the β3-N265M mutation in a mixed background were created by crossing homozygous β3-N265M (129X1/SvJ) mice [20] with WT C57BL/6J mice (Jackson Laboratories). Their heterozygous mutant offspring were bred together to create the experimental mice that we studied (i.e., β3-WT and β3-N265M homozygous mutant littermates). These breeding strategies thus produced mice carrying either the β2-N265M or β3-N265M mutations in similar mixed backgrounds (50% 129X1Sv/J, 50% C57BL/6J).

In an attempt to generate doubly mutant mice carrying both the β2-N265M and the β3-N265M mutations, we crossed homozygous mutant mice from each of the lines, but none of the 150 offspring that survived from 6 litters of two different doubly heterozygous breeding pairs was doubly homozygous for the mutation. (We would have expected 1 in 16 offspring to be doubly homozygous for the mutation.) Thus, it appears that the double mutation is lethal. This lethality presumably reflects some essential developmental contribution(s) made by these receptors that is compromised by either their functional state or their pharmacological sensitivity, not just by their presence or absence. Indeed, GABA_A_Rs are found in many organs throughout the body [51], and their critical involvement in development is exemplified by the multiple developmental abnormalities present in Gabrb3-KO mice [52].

### 4.2. Genotyping

Tail samples were acquired from each mouse and genotyped either in-house using traditional, gel-based PCR methods, or sent to Transnetyx (Cordova, TN, USA), which uses a TaqMan-based assay to collect real-time PCR data. For in-house PCR, primers were purchased from IDT (Integrated DNA Technologies, Coralville, IA, USA). The primers used for in-house PCR were as follows: β2, 5′-AGGAAGGGTCACTAGGCAGA-3′ and 5′-TTGACATCCAGGCGCATCTT-3′; β3, 5′-GTTCAGCTTCCATTCTCACTG-3′ and 5′-GTTCAGCTTCCATTCTCACTG-3′. For the β2 line, the amplified DNA was digested using PagI. Samples sent to Transnetyx and genotyped using real time PCR amplification used the following primer sequences: β2, 5′-TTTTTTCAGGAATTACAACTGTCCTAACAATG-3′ and 5′-GCACCCCATTAGGTACATGTCAAT-3′; β3, 5′-CCACCGTGCTCACCATGA-3′ and 5′-TCGATGGCTTTGACATAGGGAATTT-3′.

### 4.3. Behavioral Studies

Behavioral studies were conducted at the Waisman Center Rodent Models Core facility at the University of Wisconsin–Madison. Mice were transferred from the primary animal care facility in which they were bred and raised to the Waisman Center animal care unit at least one week prior to initiating behavioral experiments. Studies of contextual fear conditioning were carried out first, followed by elevated plus maze, and then thermal sensitivity, with 3–4 days between experiments. Open field tests were conducted on a separate cohort of mice used exclusively for those studies.

#### 4.3.1. Context Preexposure Facilitation Effect (CPFE)

To study hippocampus-dependent learning, we used a preexposure-dependent contextual fear conditioning paradigm adapted from Fanselow (1990) [53]. This paradigm, usually referred to in the literature as the Context Preexposure Facilitation Effect (CPFE) [54,55,56], takes advantage of the so-called “immediate shock deficit”, wherein animals that are shocked immediately (within several seconds) upon entry into a novel environment do not freeze on subsequent re-exposure, whereas mice that had been exposed on a prior day do exhibit a freezing response [53]. The underlying difference is thought to be that pre-exposed mice have already established a hippocampus-dependent representation of the environment (i.e., contextual memory), which takes several minutes, and that pre-formed memory can be recalled rapidly on re-exposure for subsequent association with the aversive stimulus [57,58].

The CPFE experiment took place over two weeks. During the first week, the mice were habituated and handled in the behavioral testing room for 10 min a day. During the second week, the mice underwent three experimental phases—context preexposure, conditioning, and recall—on three consecutive days. Mice were brought into the behavioral testing room 30 min prior to the testing procedure. On day 1 (context preexposure), mice were placed into the test chamber for 10 min, then returned to their home cage. The test chamber was 20 cm × 20 cm × 30 cm high, constructed of clear acrylic, with a shock grid floor consisting of stainless-steel bars 2 cm apart and a diameter of 2 mm. On day 2 (conditioning), mice were placed into the same test chamber, and after 15 s, they were administered a single foot-shock (2 s, 1 mA). The mice remained in the test chamber for an additional 30 s (47 s total time), and they were then returned to their home cage. On day 3 (recall), mice were placed back in the test chamber, and their movement was recorded using FreezeFrame^TM^ software. The percentage of time they were immobile over the first three minutes (‘freezing behavior’) served as a quantitative measure of fear memory.

#### 4.3.2. Elevated Plus Maze (EPM) Test

Mice were placed in the center of a plus-shaped maze composed of two “open” arms without walls (30 cm L × 5 cm W) and two closed arms (30 cm L × 5 cm W) enclosed by walls (10 cm H) arranged around a center zone (5 cm L × 5 cm W). Over the course of five minutes, the amount of time they spent on the arms and the number of entries of each arm were manually recorded by an experimenter blind to the genotype of the mice. The percent time the animal spent on the open arms and the number of entries to each arm served as a measure of anxiety-like behavior, with more entries and more time spent on the open arms indicating less anxiety.

#### 4.3.3. Hotplate Test

An electronically controlled hotplate (30 cm L × 30 cm W) heated to 55 C was used to measure sensitivity to a noxious thermal stimulus. The hotplate was turned on 30 min prior to testing to ensure that the desired temperature was reached. Mice were then placed individually on the hotplate, and the latency to elicit a nocifensive behavior (e.g., hind paw withdrawal or licking) was manually recorded by the experimenter blinded to the genotype of mice.

#### 4.3.4. Open Field Test (OFT)

Mice were individually placed in a square chamber (40 cm L × 40 cm W × 30 cm H) and allowed to move freely for 10 min. Mice were then injected with etomidate (10 mg/kg, 15 mg/kg, and 20 mg/kg *i.p.*) and then placed back into the chamber for another 50 min. The total distance that each mouse traveled was measured using Fusion^TM^ software (Omnitech Electronics Inc., Columbus, OH, USA).

### 4.4. Recombinant Receptor Recordings

Human Embryonic Kidney 293 cells (HEK293) were cultured in Minimum Essential Medium with Earle’s salt and L-Glutamine (Thermo Fisher Scientific, Waltham, MA, USA) supplemented with 10% fetal bovine serum (Atlanta Biologics, Flowery Branch, GA, USA) and Penicillin/Streptomycin (Lonza, Walkersville, MD, USA) and maintained at 37 °C with 5% CO_2_. HEK293 cells were transfected with cDNA encoding α5β2γ2L or α5β3γ2L (β2- or β3-WT) or α5β2(N265M)γ2L or α5β3(N265M)γ2L at a ratio of 1α:1β:3γ, with eGFP co-transfected to identify cells with successful transfection. The transfected cells were cultured for 24 h before being plated onto 12 mm glass coverslips and replaced with fresh culture medium. Patch-clamp recordings were performed 24–48 h later.

Borosilicate glass recording pipettes were pulled using a multistage micropipette puller (Flaming-Brown model P-1000; Sutter Instruments, Novato, CA, USA), fire-polished to an open tip resistance of 3–8 MΩ, and filled with an intracellular solution containing (in mM): 130 Potassium D-gluconate, 5 EGTA, 10 HEPES, 1 MgCl_2_, 5 MgATP, and 3 NaCl, pH 7.3. All cells were perfused with an extracellular solution containing (in mM): 145 NaCl, 2.5 KCl, 1 MgCl_2_, 2 CaCl_2_, and 10 HEPES, pH 7.3 (300–305 mOsm). Recordings were performed at room temperature using standard outside-out patch clamp techniques. A brief pulse of GABA (10 mM, 10 ms) in the presence or absence of 1 µM etomidate in both the ‘control’ and ‘drug’ barrels was applied to outside-out patches via a multi-barrel pipette mounted on a piezoelectric stage. Currents were low-pass-filtered at 5 kHz with an eight-pole Bessel filter, with the data collected at 20 kHz using a Digidata 1440A and Axopatch 200B amplifier (Axon Instruments, Sunnyvale, CA, USA) controlled by Clampex (ver. 10.4.1.10; Molecular Devices, Sunnyvale, CA, USA). Five to ten individual responses were averaged and analyzed using Clampfit (Axon Instruments, Sunnyvale, CA, USA). Activation and deactivation kinetics were characterized by 10–90% rise time and a biexponential decay function, respectively.

### 4.5. Data Presentation and Statistical Analysis

All values are expressed as mean ± SEM. Outliers were identified using the InterQuartile Range test using an online software package (https://www.statskingdom.com/outlier-calculator.html; k = 1.5, accessed on 10 September 2020) and excluded from further analysis. Statistical comparisons were performed using the Mann–Whitney U test for behavioral studies and two-tailed Student’s *t*-test for *in vitro* studies of receptor kinetics unless indicated otherwise. In addition, *p*-values at or below 0.05 were deemed significant. Behavioral data are presented using estimation graphics [59,60] as Gardner–Altman plots. The 95% confidence intervals for median differences were derived from bootstrap sampling distributions using an online software package (http://estimationstats.com/ accessed on 5 October 2020). Confidence intervals that included a median difference of “0” were interpreted as an indication that the two groups under comparison were similar. 

## Figures and Tables

**Figure 1 ijms-24-05637-f001:**
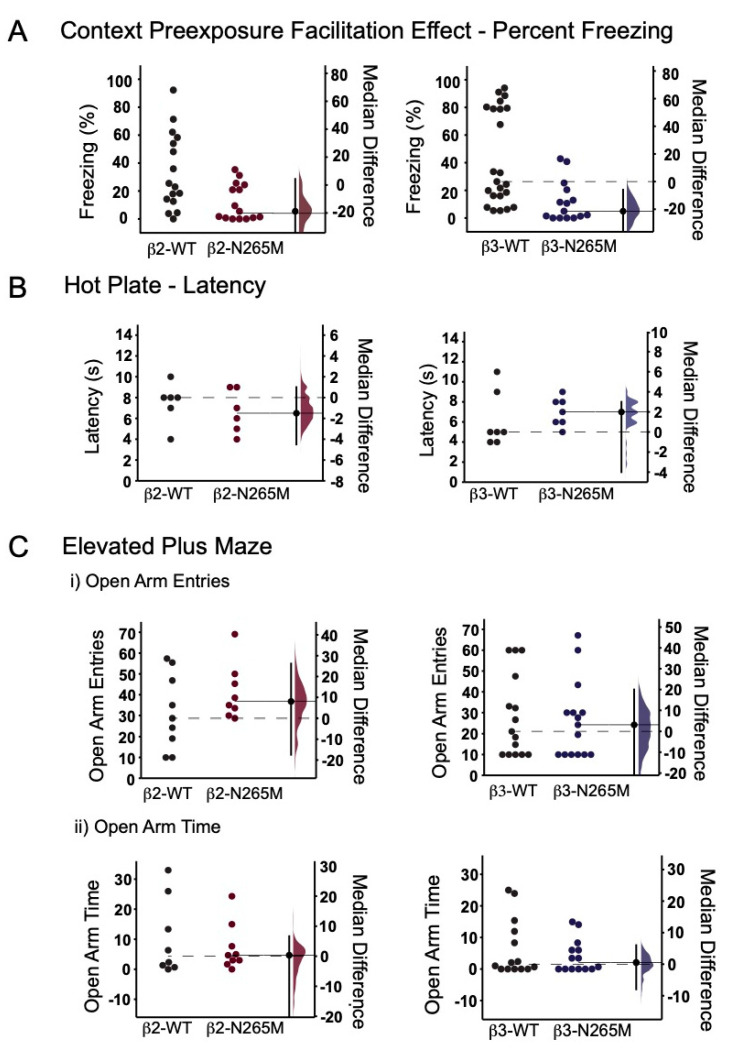
Behavioral characterization of β2-N265M and β3-N265M mice under drug-free conditions. Data are presented using Gardner-Altman estimation plots; the left axis applies to individual observations, and the right axis applies to median differences and its bootstrap sampling distribution. The median distance is shown as a large dot; the 95% confidence interval for the median difference is indicated by the ends of the vertical error bar, and in the text below as [min, max]. (**A**) Percentage of time that β2-N265M (left) and β3-N265M (right) spent freezing during the day 3 memory test. Both the β2-N265M and β3-N265M mice showed impaired memory compared to their WT counterparts (*p* = 0.04 and *p* = 0.002, respectively, Mann–Whitney U test). (**B**) Latency to nocifensive movement was similar for β2-WT vs. β2-N265M mice [−4.5, 1.0] and for β3-WT vs. β3-N265M mice [−4.0, 3.0]. (**C**) (i) Percentage of open arm entries on elevated plus maze was similar for the β2-WT vs. β2-N265M mice (left) [−17.5, 26.2] and for β3-WT vs. β3-N265M mice (right) [−22.2, 20.0]. (ii) Percent of time spent in open arms of elevated plus maze was similar for β2-WT vs. β2-N265M mice (left) [−21.0, 6.67] and for β3-WT vs. β3-N265M mice (right) [−7.97, 6.01].

**Figure 2 ijms-24-05637-f002:**
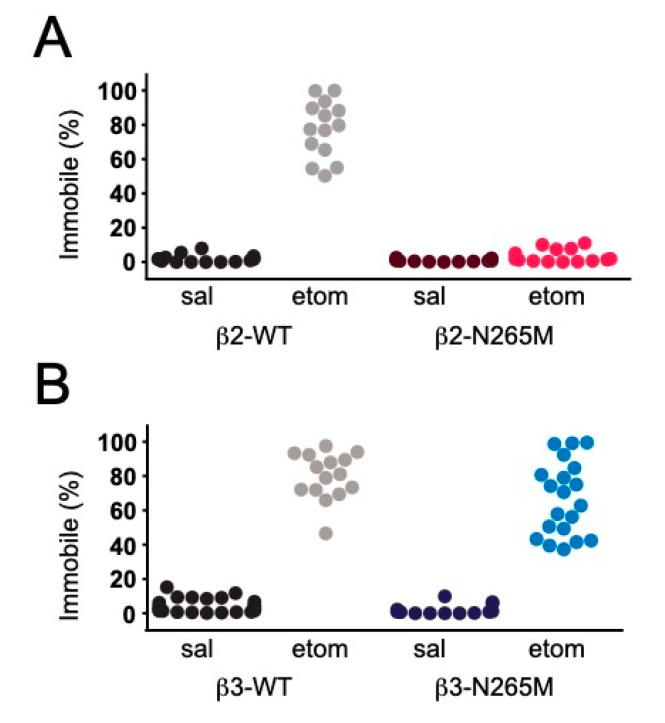
Sedative effect of etomidate. (**A**) Etomidate produced strong sedation (a high level of immobility) in β2-WT mice [95% C.I., 63.6, 87.9] but had little effect in β2-N265M mice [−0.315, 6.44]. (**B**) Etomidate produced strong sedation in both β3-WT [67.6, 87.7] and β3-N265M mice [42.6, 78.2].

**Figure 3 ijms-24-05637-f003:**
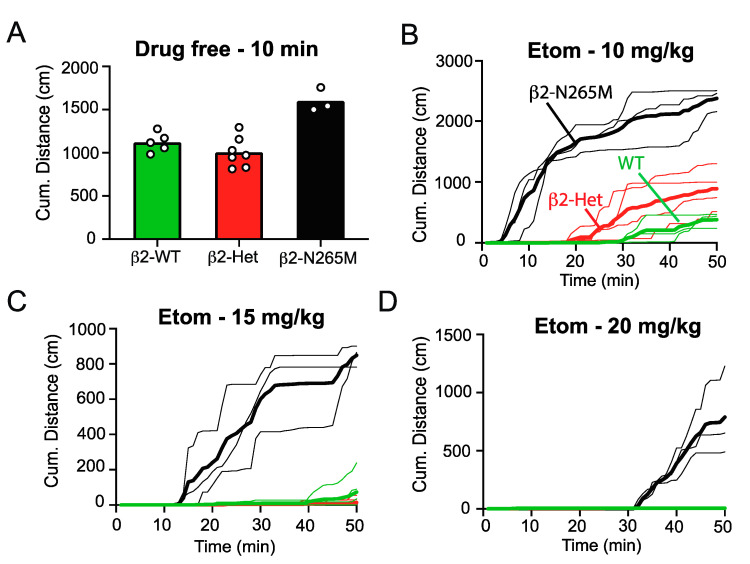
Recovery of β2-N265M mice from different doses of etomidate. (**A**) Under drug-free conditions, β2-N265M mice had a higher average cumulative distance traveled over 10 min compared to both the β2-WT and the β2-Het mice (1601 ± 136.5 vs. 1122 ± 111.7 vs. 1007 ± 172.7, respectively, *p* < 0.001, one-way ANOVA). (**B**) Following etomidate 10 mg/kg i.p., both β2-Het and β-WT mice were sedated for an average of 20 and 30 min, respectively (n = 4 and 3, respectively). β2-N265M mice resisted the sedative effect of etomidate at this dose and were sedated for an average of only 5 min (n = 3). (**C**) Following etomidate 15 mg/kg i.p., both β2-Het and β-WT mice were sedated for an average of 40 min (n = 6 and 5, respectively). β2-N265M mice resisted the sedative effect of etomidate at this dose and were sedated for an average of only 12 min (n = 3). (**D**) Following etomidate 20 mg/kg i.p., neither the β2-Het nor the β-WT mice recovered from sedation within the 50 min experiment (n = 5 and 7, respectively). β2-N265M mice were sedated for an average of 31 min at this dose (n = 3).

**Figure 4 ijms-24-05637-f004:**
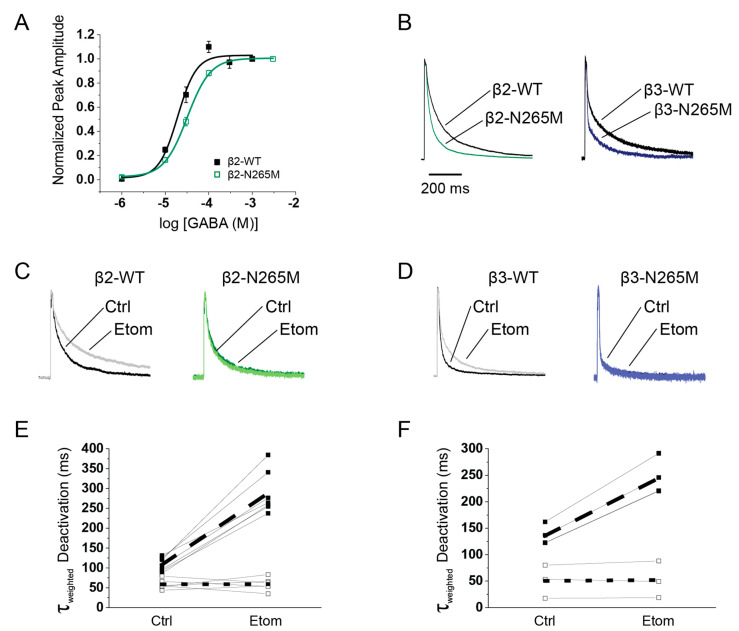
The N265M mutation altered intrinsic receptor characteristics and eliminated etomidate modulation. Recombinant α5β2γ2L, α5β3γ2L, α5β2(N265M)γ2L, and α5β3(N265M)γ2L receptors were expressed in HEK292 cells. Responses to agonist application were measured using patch clamp recording techniques. A multi-barrel application system was used to apply pulses of GABA in the presence and absence of etomidate. (**A**) The β2-N265M mutation caused a rightward shift in the GABA concentration-response curve under drug-free conditions. Peak responses were normalized to the response to 3 mM GABA. (**B**) Examples of individual responses of α5β2γ2L and α5β2(N265M)γ2L (left) and α5β3γ2L and α5β3(N265M)γ2L (right) to brief pulses of 1 mM GABA. Traces were normalized to peak current amplitude. (**C**,**D**) Examples of individual responses to α5β2γ2L and α5β2(N265M)γ2L (left) and α5β3γ2L and α5β3(N265M)γ2L (right) to application of a 10 ms pulse of 1 mM GABA, in the absence (Ctrl) and presence (Etom) of 1 μM etomidate. Responses were normalized to peak current amplitude. (**E**) 1 μM etomidate slowed deactivation of α5β2γ2L receptors (black squares, dashed line; *p* < 0.001), but not α5β2(N265M)γ2L receptors (open circles, dotted line; *p* = 0.47). (**F**) 1 μM of etomidate slowed deactivation of α5β3γ2L receptors (black squares, dashed line; *p* < 0.001), but not α5β3(N265M)γ2L receptors (open circles, dotted line; *p* = 0.93).

**Table 1 ijms-24-05637-t001:** Activation and Deactivation Kinetics in GABA-stimulated Wild-Type and β2/3-N265M Receptors.

	Receptor			
Current Phase	α_5_β_2_γ_2L_	α_5_β_2_(N265M)γ_2L_	α_5_β_3_γ_2L_	α_5_β_3_(N265M)γ_2L_
10–90% Activation (ms)	1.20 ± 0.13	1.6 ± 0.7 *	1.66 ± 0.17	2.08 ± 0.28
τ_fast_ Deactivation (ms)	41 ± 3.4	22 ± 2.1 ***	21.0 ± 6.4	3.73 ± 0.42 ***
Fraction Fast	0.60 ± 0.02	0.68 ± 0.02 *	0.60 ± 0.08	0.67 ± 0.07
τ_slow_ Deactivation (ms)	211 ± 22	137 ± 14 *	225 ± 33	91.4 ± 12.8 **
Fraction Slow	0.4 ± 0.02	0.32 ± 0.02 *	0.4 ± 0.08	0.33 ± 0.07
τ_weighted_ Deactivation (ms)	107 ± 9.6	60 ± 7.0 **	114 ± 27	36.7 ± 10.0 *
τ_weighted_ with 1 μM etomidate	285 ± 42	71.4 ± 9.6 ***	214 ± 17	37.7 ± 1.9 ***

Values represent mean ± SEM. Group sizes: β2 n = 8, β2-N265M n = 7, β3 n = 11 (etomidate n = 3), β3-N265M n = 9 (etomidate n = 3). * *p* < 0.05, ** *p* < 0.01, *** *p* < 0.001, compared to wild type.

## Data Availability

All the data generated by this study are contained within the manuscript, so they are not deposited in a public repository.

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
