# Peer review of "Changes in Memory, Sedation, and Receptor Kinetics Imparted by the β2-N265M and β3-N265M GABAA Receptor Point Mutations"

_ijms, 2023, doi:10.3390/ijms24065637_

Round 1
Reviewer 1 Report
In this very well written manuscript, the authors found that i) mice carrying the N265M mutation in either the beta2 or beta3 subunit exhibited impaired contextual conditioning, but they were unchanged in the other behavioral measures; ii) beta2-N265M mice, but not beta3-N265M mice, resisted etomidate-induced sedation; iii) the beta-N265M mutation produced a slight increase in exploratory activity under baseline conditions, but it strongly accelerated recovery from etomidate administered in a gene dose-dependent manner and iv) the beta2-N265M and beta3-N265M mutations both accelerated receptor deactivation by approximately two to three-fold following a brief pulse of 100 GABA.
1. Experiments in vitro in which etomidate blocked long term potentiation in hippocampal brain slices from beta3-N265M but not beta2-N265M mice further supported a role for beta2-GABAARs, but not beta3-GABAARs, in etomidate-induced memory suppression. But is this just their localization since the two receptors, especially at the site of mutation, are extremely homologous?
2. Can the authors postulate why the double mutant is lethal despite the minimal effects on baseline function of either mutant alone?
3. It seems that the beta 2 is required for sedation, especially that mediated by etomidate, beta 3 for movement response to stimuli, and both for the associated memory functions demonstrated here. The latter seems to require the association with the alpha 5 subunit. Can the authors comment on or identify any specific circuit components that are most influential in these processes to further shed light on localization of this functionality given such specific receptor stoichiometry?
Reviewer 2 Report
The manuscript by Abdulzahir et al further characterizes mice with point mutations in the beta2 or beta3 subunit of their GABAA receptors. These mutants have been important for dissecting the molecular targets by which general anesthetics exert distinct effects on memory and sedation. In this study, the authors report several new findings regarding pharmacological and behavioral similarities and differences between wild-type mice and the two mutants, and reproduce a previous result in which one of the mutants exhibits strong but not total resistance to etomidate anesthesia.
The results in the absence of anesthetic drug show memory impairments in both mutants relative to wild-type animals, but no differences in anxiety or pain sensitivity. These results are important because they constrain the interpretation of the results characterizing the effects of anesthetics on the mutants. Moreover, the results showing resistance of the beta2 mutant to etomidate anesthesia is also significant because the present study used a different background strain of mouse, supporting that the resistance conferred by the mutation is robust to variations in genetic background. The study also reports in vitro results showing that both mutations accelerate deactivation of GABAA receptors, contributing to the molecular characterization of the functional consequences of each mutation.
Below I list a number of suggestions for corrections or clarifications to the manuscript.
Introduction:
37-38. Do these references establish that GABAA receptors are a significant functional target of volatile anesthetics as well as injectables? If not the sentence/paragraph could be sharpened so that it only refers to anesthetics for which GABA receptors are known to play a significant role. This suggestions might also apply in the discussion.
55-61. In these lines sedation is attributed to alpha5 and beta2 receptors, while LORR is attributed to beta3 receptors. This is a bit confusing as LORR results from sedation and hypnosis in my current understanding. If this could be clarified that would be desirable, but it may be that this is the way the results were reported by other authors.
Results:
107-108. The acronyms CPFE, OFT, and EPM are used before they are defined.
Figure 1 caption:
137-138. The caption refers to top and bottom panels of Figure 1A, which do not exist.
Discussion:
The result that both mutations lead to comparable changes in receptor kinetics raises the question: why don’t both mutants confer resistance to sedation? This question should be addressed in the Discussion.
These are all my comments; I am not submitting any secret comments to the editor.
